# Strategies to assess and promote the socio-emotional competencies of university students in the socio-educational and healthcare fields: A scoping review

Natalia Gandía-Carbonell[1]*, Cristian Molla-Esparza[2], Sònia Lorente[1], Paz Viguer[3], Josep-Maria Losilla[1]

1 Department of Psychobiology and Methodology of Health Sciences, Faculty of Psychology, Autonomous University of Barcelona, Bellaterra, Barcelona, Spain, 2 Department of Research Methods and Diagnostics in Education, Faculty of Philosophy and Education Sciences, University of Valencia, Valencia, Spain, 3 Department of Developmental and Educational Psychology, Faculty of Psychology, University of Valencia, Valencia, Spain

* natalia.gandia@autonoma.cat

## Abstract

This scoping review systematically analyses and synthesises empirical evidence on measures and intervention programmes aimed at promoting socio-emotional competencies (SECs) in university students in socio-educational and healthcare fields. A comprehensive literature search was conducted of the Scopus, PubMed, ERIC and PsycINFO databases, and a narrative synthesis approach was employed to analyse the findings from a total of 288 studies. The results highlight a general consensus on the importance of fostering SECs in university students in both fields, while revealing a significant lack of research in the socio-educational sector. Regarding both populations, a notable heterogeneity was found in the measurement of SECs and in the wide variety of tools used, which were based on different theoretical approaches, and were often not standardised or not exclusively designed to measure this type of competencies. In the intervention programmes reviewed, the SECs most frequently promoted were empathy and interpersonal emotional perception, communication, and identification, understanding, and regulation of one's own emotions. Nonetheless, many studies lacked detailed reporting on the theoretical frameworks and intervention procedures applied, therefore limiting their replicability. Future intervention programmes should align targeted competencies with students' profiles, future roles and professional needs, using standardised, profile-adapted measures to better evaluate their effectiveness.

**Data availability statement:** The data supporting the findings of this study are openly available in the Open Science Framework repository at https://osf.io/n5y6j.

**Funding:** The research is supported by Grant PID2022-141403NB-I00, from the Spanish Ministry of Science and Innovation. The authors declare no conflict of interest, and the funders had no role in the study design, data collection, analysis, decision to publish, or preparation of the manuscript. No additional external funding was received for this study.

**Competing interests:** The authors have declared that no competing interests exist.

## 1. Introduction

Professionals of socio-educational fields (e.g., social workers, social educators) and healthcare fields (e.g., doctors, nurses, psychologists) make up a wide array of roles involving the provision of care services towards people who are in delicate or vulnerable moments due to their physical, social and/or emotional needs [1]. Such professionals regularly face complex situations that demand motivation, vocation and emotional involvement [2–5]. Therefore, it is evident that their university training should go beyond imparting knowledge, technical expertise and skills, to also focus on the development of socio-emotional competencies (SECs for short) for professional performance and personal well-being [6–9].

SECs are a set of knowledge, skills, attitudes, and dispositions that are essential for effectively understanding, expressing, and regulating emotions, managing behaviour, and building positive relationships [10–12]. Although SECs represent a construct with different approaches, definitions, and classifications [13], there is a general consensus that a core set of SECs is essential and can be developed and acquired through learning [14] (S1 Table summarizes the key theoretical models and their respective SECs).

Enhancing university training in SECs for future professionals can improve their ability to effectively address the needs and care of patients (as well as clients or users), ultimately increasing satisfaction with healthcare services. Likewise, training professionals qualified in certain SECs may enhance their well-being, which might otherwise be eroded by daily work demands. In terms of professional performance, the promotion of SECs can lead to greater effectiveness in verbal and non-verbal communications [15,16], an enhanced capacity for conflict management [17], a facilitated building of therapeutic relationships [18], and enhanced empathy in understanding and responding appropriately to the needs of service users [19]. Moreover, when healthcare professionals can better understand and manage both the emotions of their patients and their teammates, significant improvements are seen in satisfaction with the patient experience and the care provided [1,20], in more effective interactions between professionals, in better decision-making, and in the development of critical thinking [21].

Concerning personal benefits, SECs can contribute to better stress management [15], improvements in mental health [22], the prevention of burnout, and the maintenance of psychological well-being over the duration of one's professional career [2,23,24]. As individuals become more proficient in managing their own emotions, they gain personal confidence to navigate complex interpersonal dynamics and challenges [19], enhancing their self-efficacy and positively impacting their professional performance [25].

Despite the clear benefits of promoting SECs of professionals of healthcare and socio-educational fields, comprehensive evaluations of empirical evidence on how these competencies can be effectively assessed and promoted remain limited. To date, existing literature reviews have been generally limited to the university healthcare field, and focused almost entirely on nursing and medicine degrees [7,26–28]. To the best of our knowledge, no reviews have specifically addressed the

assessment and promotion of SECs in the university socio-educational field. Our review thus aims to bridge this gap by providing a comprehensive examination of both socio-educational and healthcare fields, extended to include other related and relevant disciplines (e.g., Psychology, Physiotherapy). Other reviews have not only focused on specific disciplines, but also on examining associative objectives between SECs and other variables [29], or on evaluating specific intervention results in relation to particular SECs, such as empathy [30–34]. While these reviews have provided valuable insights, they have been limited in scope.

To overcome such research limitations, we considered it imperative to carry out a scoping review to identify, map, analyse and synthesise the existing empirical evidence on strategies proposed to assess and promote university students' SECs in both the socio-educational and healthcare fields. A scoping review is the type of systematized review (i.e., systematic, transparent and replicable) most appropriate to address this type of objectives [35]. The two specific objectives of our scoping review were: 1) to identify the most frequently used measures to assess university students' SECs, the theoretical frameworks of these measures, and their quality; and 2) to identify and describe the intervention programmes proposed to date for the promotion of SECs at the university stage, their theoretical frameworks, the specific SECs promoted, and their outcomes. Ultimately, this research aims to contribute to the training in SECs of future health and socio-educational professionals, providing valuable information to improve the integration of these competencies into university curricula.

## 2. Methodology

### 2.1. Protocol and registration

This scoping review was undertaken in accordance with the PRISMA Extension for Scoping Reviews (PRISMA-ScR) recommendations [36], and with a previously published protocol article [13]. A synthetic protocol for this scoping review has been registered in the INPLASY repository (number 202.120.076; https://doi.org/10.37766/inplasy2021.2.0076). In accordance with the principles of open science, all datasets and supplementary information referenced in the following sections are publicly available in the OSF repository and can be accessed at https://osf.io/n5y6j. The PRISMA-ScR Checklist applied can be consulted in the S1 Checklist.

### 2.2. Data sources

A literature search was performed in three major literature databases for the socio-educational and healthcare fields: PsycINFO via EBSCOHost, Education Resources Information Center (ERIC) via ProQuest, and PubMed via the National Center for Biotechnology Information (NCBI). The multidisciplinary bibliographic database Scopus (via Elsevier) was also scrutinised to maximise the exhaustiveness of the search. Additionally, a grey literature search was conducted using the Google search engine, and reviewing the first 150 entries sorted by relevance, to minimise potential publication bias. The bibliographic references of selected articles were also checked to retrieve other potentially eligible studies. Finally, prominent researchers on this topic were questioned about ongoing and currently unpublished studies.

### 2.3. Search strategy

The search strategy followed the Peer Review of Electronic Search Strategies (PRESS) guidelines [37]. The search strategy was developed through an iterative process of consultation of key terms from empirical literature, database searches, and previously published reviews and protocols. There were therefore two main groups of terms. The first group referred to SECs, including descriptors related to the primary theoretical approaches. The second group focused on the target population of university students. As far as possible, searches were designed using combinations of free text and thesaurus terms. The search was limited to publications from the past two decades, specifically from 2000 to

26 January 2024, and search alerts were set across all the data bases. Results were further refined by research design (empirical studies) and publication language (English and Spanish). The retrieved citations were exported into the Mendeley Reference Manager v2.119. To facilitate the replication of this review, the full search strategy for each database is provided in the S2 Table.

### 2.4. Study eligibility criteria

In accordance with the research objectives, studies were included if they: a) were experimental studies (randomised or non-randomised), non-experimental (cohort, case-control or cross-sectional), single-case or qualitative designs; b) studied a sample of undergraduate, graduate or postgraduate university students being trained to become professionals in socio-educational and healthcare fields; and c) provided sufficient information on the strategies used to assess or promote university students' SECs.

Theoretical and case report studies were excluded, along with those whose objectives were exclusively academic reviews or evaluations of university study plans in socio-educational and healthcare fields.

### 2.5. Study selection

Duplicated references were removed prior to the screening of studies using Mendeley Reference Manager v2.119. The study eligibility process was independently conducted by two researchers (NGC & PV). In a first step, the literature was checked by title and abstract according to the inclusion and exclusion criteria. When decisions could not be made using the title and abstract alone, the full paper was retrieved. In a second step, the researchers examined the full text of the preselected studies for inclusion in the review. Any disagreement between the reviewers was resolved through discussion with a third researcher (JML). Agreement between the reviewers during the study selection process was analysed using Cohen's kappa [38], calculated to be.83 (95% CI:.76,.96; with 93% observed agreement). All the studies finally included in the review are listed in the S1 Text.

### 2.6. Data extraction

Data extraction was conducted according to a predefined coding protocol (see S3 Table). Data were extracted independently by two researchers (NGC, CME, or SL), using an *ad hoc* checklist to report: a) identification data (e.g., authors, publication year); b) sample characteristics (e.g., sample size, gender distribution); c) methodological data (e.g., research design, sampling technique); and d) substantive data (e.g., theoretical SECs framework, measures of SECs). Discrepancies were resolved through discussion with a third researcher (JML or PV). Agreement between reviewers was analysed using Cohen's kappa [38], calculated to be.76 (95% CI:.63,.88; with 88% observed agreement).

### 2.7. Data synthesis strategy

Results were presented using a narrative synthesis approach. To map the existing evidence, the studies were classified based on their objective: a) measure studies (i.e., studies aimed at developing, adapting and/or testing SECs measures); b) descriptive and correlational studies (i.e., studies with the aim of describing or examining associations between SECs and other variables); and c) intervention studies (i.e., studies specifically tailored to evaluate intervention programmes for the promotion of SECs). A detailed descriptive analysis was developed of the target population, identifying geographical origin, gender, age, field of study field, and educational level. Regarding measure studies, data were synthesised on the SECs assessed, their theoretical frameworks and their quality indicators. Finally, intervention studies were classified according to their main characteristics, including theoretical SECs frameworks of reference, the procedures used, SECs promoted, and their efficacy and effectiveness. The analysis of sample representativeness and procedure replicability was also reported.

## 3. Results

### 3.1. Study selection results

Fig 1 summarises the identification and selection process for the 288 studies included in the review. Of these, 23 were measure studies that developed or tested measures of SECs, 201 were descriptive and correlational studies that analysed associations with SECs, and 64 were intervention studies aimed at promoting or improving SECs. We note that almost half of the included studies (n=142; 49%) were published from 2019 to 2023.

### 3.2. Sample characteristics and research designs

All of the measure studies exclusively regarded the healthcare field (n=22; 96%), except for one that combined the healthcare with the socio-educational field. The age of sample participants in the studies ranged from 17 to 79 years. The predominant educational level was undergraduate (n=22; 96%), and the degrees that stood out were Nursing (n=13; 56%), Psychology (n=7; 30%), and Medicine (n=6; 26%). These studies mostly used cross-sectional designs (n=23), with non-probabilistic samples (n=22; 96%). A significant portion (n=11; 48%) aimed to report the psychometric properties of existing measures, while others developed new measures (n=9; 39%) or adapted existing ones (n=3; 13%) (supplementary Dataset 1).

The descriptive and correlational studies were conducted in 47 different countries. The most represented country was the United States (n=40; 20%), followed by Spain (n=25; 12%), and Iran (n=14; 7%). As in the measure studies,

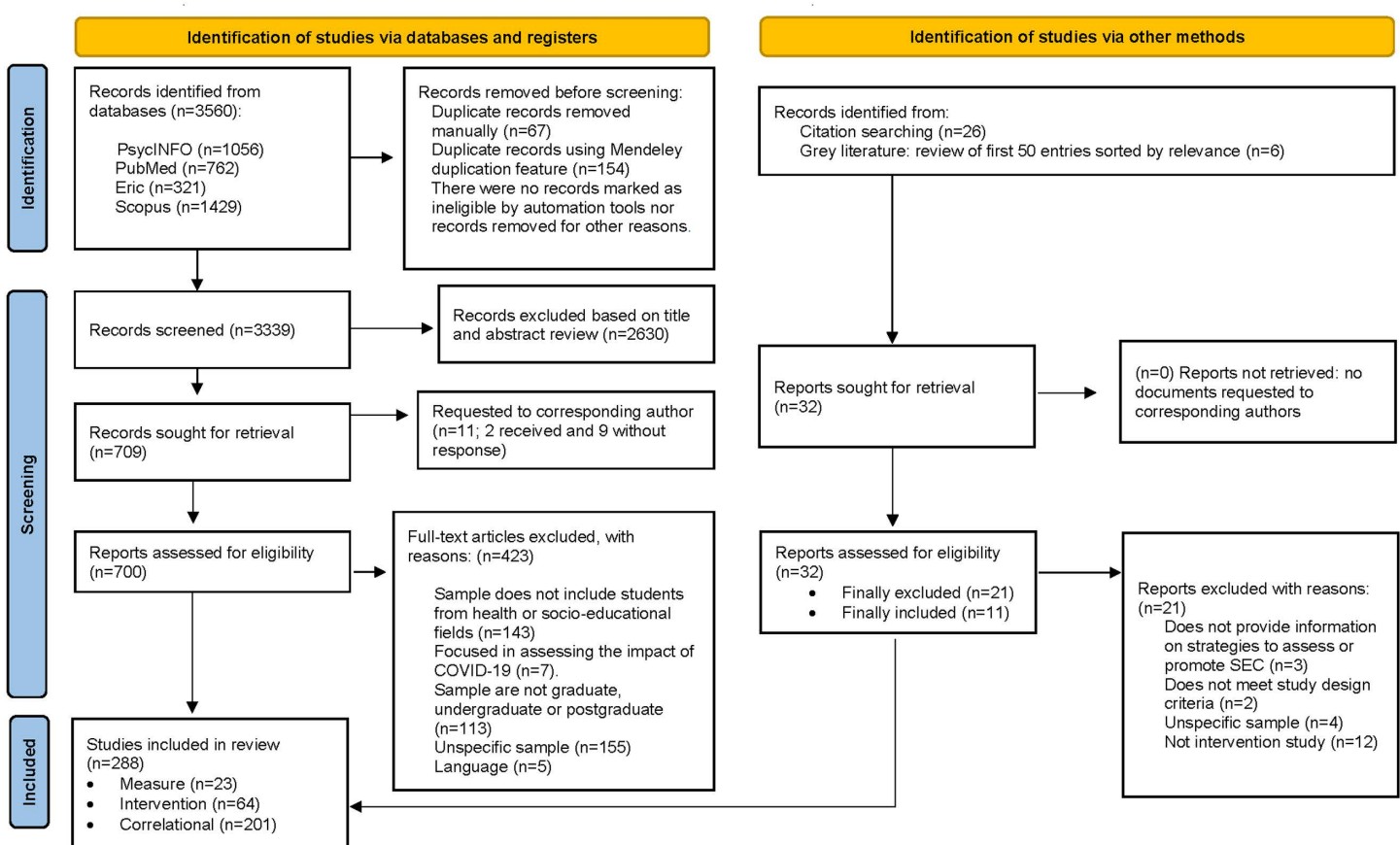

**Fig 1. PRISMA 2020 flowchart of the scientific literature search and selection.** *Note.* Figure adapted from Page et al. [39].

the majority of these studies focused exclusively on the healthcare field (n = 162; 80%), while a minority focused on the socio-educational field (n = 7; 3%). A few studies (n = 7; 3%) combined the socio-educational and healthcare fields, and a smaller percentage combined these two fields with others (n = 11; 5%). The age of sample participants ranged from 16 to 65 years. In general, the predominant educational level was undergraduate (n = 156; 78%), followed by postgraduate (n = 24; 12%). In the healthcare field, studies predominantly focused on Nursing (n = 67; 33%), Medicine (n = 58; 29%), and Psychology (n = 31; 15%). In the socio-educational field, among the studies that reported specific degree information, the most notable degrees were in Social Education (n = 7; 3%) and Social Work (n = 7; 3%). Cross-sectional designs were predominant (n = 164; 82%), followed by longitudinal approaches (n = 24; 12%). Most studies employed non-probabilistic sampling (n = 173; 86%), with only 9% (n = 19) using probabilistic samples (supplementary Dataset 2).

Finally, the intervention studies were also conducted predominantly in the United States (n = 24; 38%) and Spain (n = 6; 9%). Sample participants ages spanning from 17 to 68 years, with the majority falling within the 18–29 age group. Most studies (n = 57; 89%) were developed in the healthcare field, mainly in Medicine, Nursing and Psychology, and only 14% (n = 9) in the socio-educational field, with 3% (n = 2) of the studies combining both fields (see Fig 2; for further information, see the supplementary Dataset 3).

Intervention studies showed greater variability in their research designs, with non-randomised approaches being the predominant choice (n = 32; 50%), followed by randomised designs (n = 13; 20%). A minority of studies (n = 9; 14%) adopted qualitative methodologies. The rest of the studies used mixed-methods (n = 8; 13%) and pre-experimental (n = 2; 3%) designs. Regarding sampling methods, the majority used non-probabilistic sampling (n = 62; 97%), with only two studies using probabilistic sampling.

### 3.3. SECs measures applied in university students

Table 1 shows the most commonly used specific standardised measures. The full characteristics of all the measures identified in this review can be consulted in the supplementary Dataset 4. Three types of standardised measures were identified: 1) specific measures, developed to specifically assess SECs (e.g., SSREI) [40]; 2) non-specific measures, designed

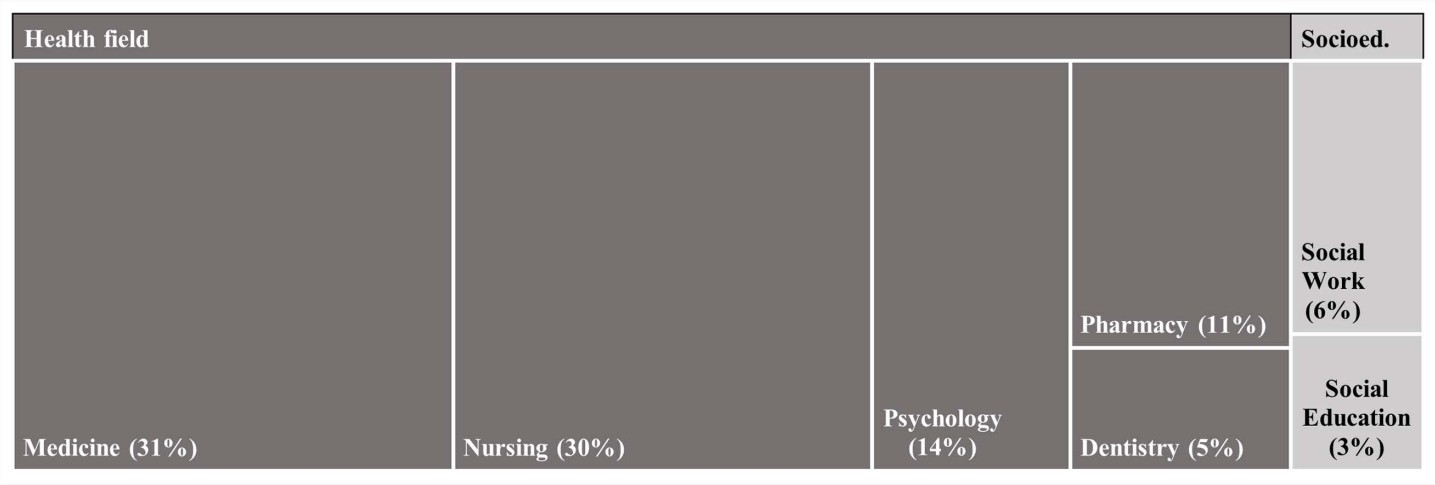

**Fig 2. University degrees of the students sampled in the intervention studies (n = 64).** *Legend.* Socioed. = Socio-educational field. The figure shows degree programmes with counts greater than 1, or representing more than 2%. The degree programmes not included in the figure belonging to the health field are: Emergency Medicine, Gynaecology, Health Psychology, Surgery, Nutrition, Obstetrics, Occupational Therapy, Physiotherapy, Psychological Counselling and Guidance, Public Health Practitioners, Radiology, and Radiotherapy, each representing 2%.

**Table 1. Most used specific standardised measures.**

| Measure (last version) | Versions used | Translations | SECs assessed | Theoretical framework | Studies that used it (year range) | Studies assessing reliability/ validity |
|---|---|---|---|---|---|---|
| SSREI Schutte et al. 1998 [40] | MSSREI | 1. SSREI Farsi 2. SSREI Hebrew 3. SSREI Polish 4. SSREI Arabic 5. SSREI Slovenian 6. SSREI Chinese | Appraisal of emotions in the self \| Appraisal of emotions in others \| Emotional expression \| Emotional regulations of the self \| Emotional regulation of others \| Utilization of emotions in solving problems | Salovey & Mayer 1990 [43] | 38 (2011-2023) | 22/ 7 |
| TEIQue Petrides 2009 [44] | TEIQue-SF | 1. TEIQue French 2. TEIQue-SF French 3. TEIQue-SF Slovenian 4. TEIQue-SF Persian 5. TEIQue-SF Greek | Well-being (trait optimism, trait happiness, self-esteem) \| Self-control (emotion regulation, impulsiveness, stress management) \| Emotionality (trait empathy, emotional perception, emotional expression, relationships) \| Sociability (emotion management, assertiveness, social awareness) \| Adaptability & Self-motivation | Petrides & Furnham 2001 [45] | 30 (2010-2023) | 13/ 4 |
| MSCEIT V2.0 Mayer et al. 2003 [46] | MSCEIT V1.1 | 1. MSCEIT V2 Greek 2. MSCEIT Spanish 3. MSCEIT Japanese | Perceiving emotions \| Facilitating emotions \| Understanding emotions \| Managing emotions | Salovey & Mayer 1990 [43] | 26 (2003-2023) | 10/ 3 |
| EQ-i 2.0 Bar-On 2011 [47] | 1. EQ-i 125 items 2. EQ-i 90 items 3. EQ-i:S 51 items 4. EQ-i 133 items 5. EQ-i-M20 | 1. EQ-i 90 items Persian 2. EQ-i 2.0 Greek 3. EQ-i-M20 Peruvian 4. EQ-i 133 items Italian 5. EQ-i-M20 Spanish 6. EQ-i 88 items Turkish | Self-perception (self-regard, self-actualization and emotional self-awareness) \| Self-expression (emotional expression, assertiveness and independence) \| Interpersonal (interpersonal relationship, empathy and social responsibility) \| Decision making (problem solving, reality testing and impulse control) \| Stress management (flexibility, stress tolerance and optimism) | Bar-On 2006 [48] | 21 (2008-2022) | 7/ 2 |
| TMMS Salovey et al. 1995 [49] | TMMS-24 Adaptation | 1. TMMS-24 Spanish 2. TMMS-24 Portuguese | Attention to feelings \| Clarity of feelings \| Mood repair | Salovey & Mayer 1990 [43] | 20 (2003-2023) | 19/ 2 |
| WLEIS Wong & Law 2002 [50] | None | 1. WLEIS Chinese 2. WLEIS Spanish 3. WLEIS Korean 4. WLEIS Peruvian | Self-emotion appraisal \| Others' emotion appraisal \| Use of emotion \| Regulation of emotion | Salovey & Mayer 1990 [43] | 15 (2009-2023) | 15/ 9 |
| RSES Rosenberg 1965 [51] | None | 1. RSES Korean 2. RSES Portuguese 3. RSES Spanish | Self-esteem | Rosenberg 1965 [51] | 10 (2000-2024) | 6/ 0 |
| EQSAC Sterrett 2010 [52] | EQSAC Adaptation | None | Self- awareness \| Self-confidence \| Self-control \| Empathy \| Motivation \| Social competence | Sterrett 2010 [52] | 8 (2008-2023) | 2/ 1 |
| EIA Bradberry & Greaves 2009 [53] | None | None | Self-awareness \| Self-management \| Social awareness \| Relationship management | Bradberry & Greaves 2009 [53] | 6 (2018-2021) | 4/ 3 |
| JSPE Hojat et al. 2016 [54] | JSPE Student version | JSPE Spanish | Empathy | Non-reported | 6 (2009-2023) | 2/ 3 |

*(Continued)*

**Table 1.** (Continued)

| Measure (last version) | Versions used | Translations | SECs assessed | Theoretical framework | Studies that used it (year range) | Studies assessing reliability/ validity |
|---|---|---|---|---|---|---|
| GSES Sherer et al 1982 [55] | GSES 17 items | 1. GSES Brazilian 2. GSES Spanish | Self-efficacy | Bandura 1977 [56] | 6 (2018-2020) | 4/ 1 |
| Nicholas Hall Scale (NR) | None | Nicholas Hall Scale Turkish | Awareness of emotions \| Managing emotions \| Self-motivation \| Empathy \| Social skills | Non-reported | 6 (2018-2020) | 3/ 0 |

*Legend.* SECs = Socio-emotional competencies; EI = Emotional intelligence; n.d. = No date; NR = No reference; SSREI = Schutte Self-Report Emotional Intelligence Test; TEIQue = Trait Emotional Intelligence Questionnaire; MSCEIT = Mayer-Salovey-Caruso Emotional Intelligence Test; EQ-i = Emotional Quotient Inventory; TMMS-24 = Trait Meta-Mood Scale; WLEIS = Wong and Law Emotional Intelligence Scale; RSES = Rosenberg Self-Esteem Scale; EQSAC = Sterrett's Emotional Quotient Self- Assessment Checklist; JSPE = Jefferson Scale of Physician Empathy; GSES = General Self-Efficacy Scale; EIA = Bradberry and Greaves' Standard Emotional Intelligence Questionnaire; MSSREI = Modified Schutte Emotional Intelligence Scale; TEIQue-SF = Trait Emotional Intelligence Questionnaire Short Form.

primarily to evaluate other variables than SECs but which have also been used to assess SECs (e.g., FATCOD-S) [41]; and 3) mixed measures, combining the assessment of SECs with other variables (e.g., NCSS) [42].

A total of 121 standardised measures were identified, classified into three categories: 66 specific, 35 non-specific, and 20 mixed. As shown in Fig 3, standardised specific measures predominated in all three study types, accounting for 61% (n = 14) in measure studies, 90% (n = 180) in descriptive and correlational studies, and 63% (n = 40) in intervention studies. In the case of intervention studies, 38% (n = 24) employed standardised specific measures exclusively (i.e., not in combination with any other type of measure). We also note that 27% (n = 17) of intervention studies employed qualitative measures, 9 of which exclusively, 3 in combination with standardised specific measures, and 5 in combination with non-standardised measures.

Among the standardised measures, the Schutte Self-Report Emotional Intelligence (SSREI) scale [40] emerged as the most used specific measure in the studies included in the review (n = 39; 14%). This instrument evaluates various aspects, including the appraisal of emotions in oneself and in others, emotional expression, emotional regulation of oneself and of others, and the use of emotions in problem-solving. More than half (n = 22; 56%) of these studies assessed and reported the reliability of this instrument in relation to the study sample, and 18% (n = 7) also reported its validity.

The Trait Emotional Intelligence Questionnaire (TEIQue) [44] was the second most used standardised specific measure (n = 31; 11%), which evaluates the subdimensions of well-being, self-control, emotionality, sociability, adaptability, and self-motivation. Of the studies that used this measure, 42% (n = 13) reported reliability, and just 13% (n = 4) validity.

The third most used measure, employed by 9% (n = 27) of the studies, was the Mayer-Salovey-Caruso Emotional Intelligence Test (MSCEIT) [46], which assesses competencies in perceiving, facilitating, understanding and managing emotions. Of the studies that used this measure, 41% (n = 11) reported reliability, and just 11% (n = 3) validity.

Other measures that stood out among the most used were the Emotional Quotient Inventory (EQ-i) measure [57] (n = 22; 8%), the Trait Meta-Mood Scale (TMMS) [49] (n = 22; 8%), and the Wong and Law Emotional Intelligence Scale (WLEIS) [50] (n = 17; 6%).

The SECs assessed varied according to the measures used and their underlying theoretical frameworks. All of the most commonly used measures are based on the theoretical construct of emotional intelligence (EI for short), although they draw on different theoretical models. On the one hand, the SSREI [40], MSCEIT [46], TMMS [49] and WLEIS [50] are based on the theoretical model of Salovey & Mayer [43]. These measures exhibit similarities in assessing SECs focused on the management of emotions, by identification, regulation and expression of both one's own emotions and those of

**Fig 3. SECs measures by study type.**

others. On the other hand, the TEIQue [44] is based on the theoretical framework of Petrides & Furnham [45], and the EQ-i [57] is based on the Bar-On model [57], with each assessing different SECs.

### 3.4. Intervention programmes to promote SECs in university students

A total of 64 studies reported on intervention programmes aimed at promoting SECs, revealing three distinct scenarios. First, 41% (n = 26) of the interventions were focused exclusively on SECs; second, 34% (n = 22) of these studies primarily assessed SECs but also examined other constructs, such as burnout, anxiety, and creative thinking; and third, 25% (n = 16) conducted only a secondary analysis of SECs. Table 2 shows the studies focused exclusively on addressing SECs, summarising information on sample characteristics, research design, theoretical framework, SECs promoted, intervention procedures, and measures used.

**3.4.1. SECs promoted in intervention programmes.** There was a lack of consensus regarding the definition and classification of SECs in the intervention studies, as well as about which SECs should be promoted among university students in socio-educational and healthcare fields. Only 31% (n = 20) of these studies reported their theoretical frameworks, identifying up to 14 different frameworks, the majority aligned with the evaluation measures employed. However, notably, 69% (n = 44) of the studies did not reference any theoretical framework to justify interventions.

Fig 4 shows the SECs addressed by the interventions, grouped into three categories: intrapersonal, interpersonal, and general SECs constructs. The specific SECs included in each category are detailed in Dataset 5, available at the OSF repository. The intrapersonal category refers to identifying, understanding, managing and expressing emotions, along with other SECs such as self-esteem, self-efficacy, and self-confidence, and other skills necessary for personal management and well-being. In contrast, the interpersonal category refers to abilities in empathizing, communicating effectively, and managing conflicts, focusing on skills necessary for successful interactions and managing relationships with others. Finally, the third category comprises those studies that did not mention specific SECs, but rather general constructs, such as EI, socio-emotional skills, and emotional competencies.

The two most promoted SECs belong to the interpersonal category. Almost half of the intervention studies (n = 31; 48%) reported interventions on empathy and interpersonal emotional perception, referring to SECs such as identifying, analysing and understanding others' emotions or social awareness. Moreover, 38% (n = 24) of the intervention studies focused on communication, addressing SECs such as listening skills and assertiveness. The next two most frequently promoted SECs belong to the intrapersonal category, with 34% (n = 22) of the intervention studies focused on identifying, understanding and facilitating one's own emotions, and 30% (n = 19) on regulating and managing them.

Finally, 23% (n = 15) of the intervention studies reported interventions on general SECs constructs, including EI, socio-emotional skills, and emotional competencies.

**3.4.2. Characteristics of intervention programmes.** Regarding the number of intervention sessions, 28% (n = 18) of the studies delivered one to five sessions, 17% (n = 11) from six to ten, and 6% (n = 4) exceeded 10 sessions. Notably, close to half of the studies (n = 31; 48%) did not specify the number of sessions delivered, and 62% (n = 40) did not report the duration of the intervention sessions. Of the intervention studies that reported session duration, 19% (n = 12) lasted less than 10 hours, 12% (n = 8) ranged from 10 to 20 hours, 3% (n = 2) from 20 to 30 hours, and 3% (n = 2) reported durations between 52 and 276 hours.

The number and periodicity of the intervention sessions also varied considerably: 8% (n = 5) were carried out on a single occasion, 3% (n = 2) on a daily basis, 34% (n = 22) on a weekly basis, 3% (n = 2) on a biweekly basis, and 6% (n = 4) on a monthly basis. Notably, 39% (n = 25) failed to provide this information.

In general, the studies did not report information on the temporal moment within the university course or curriculum, nor on prerequisites established by administrators, the context of the intervention, or the type of support material used.

**Table 2. Characteristics of intervention studies primarily targeting SECs (n = 48).**

| Study | Sample size (% fem.) | Country | Degree programme | Study design | SECs promoted | Reported SECs theoretical framework | Modality | N° of sessions (hours; frequency) | Teaching strategies | SECs measures |
|---|---|---|---|---|---|---|---|---|---|---|
| Abdulqadir et al. 2022 [58] | 247 (63%) | Iraq | Nursing | QE | Emotional intelligence-based strategies | Bar-On 2006 [48] | NR | 10 (NR; NR) | NR | EQ-i |
| Abe et al. 2013 [59] | 181 (57%) | NR | Medicine | QE | Expressing one's feelings \| Listening others' feelings | NR | EC | 1 (NR; once) | Pair work \| Group work | TEIQue |
| Antoun et al. 2020 [60] | 27 (59%) | Lebanon | Medicine | QE | Interpersonal skills | Salovey & Mayer 1990 [43] | C | 20 (NR; biweekly) | Case study \| Group work \| Debriefing | MSCEIT V2.0 |
| Beauvais et al. 2019 [61] | 79 (92%) | USA | Nursing | MM | Emotional intelligence \| Communication skills | Hildegard Peplau's 1997 [62] | C | 6 (12; weekly) | Psychodrama \| Group work \| Role playing | MSCEIT V1.1 \| Recordings |
| Bonvicini et al. 2009 [63] | 160 (36%) | USA | Medicine | QE | Empathy | NR | EC | NR (NR; monthly) | Didactic and experiential methods \| Individual coaching | ECCS \| GRS |
| Brison et al. 2015 [64] | 74 (85%) | NR | Psychology | E | Self-development \| Emotional competence \| Empathy \| Listening skills | NR | EC | 3 (12; weekly) | Person-centred group \| Role playing | TEIQue \| SI \| TCCA-B \| TICA \| ES \| OGBHS |
| Caballero-García & Sánchez-Ruiz 2021 [65] | 300 (77%) | Spain | NR | E | Emotional self-awareness \| Assertiveness \| Emotional regulation \| Empathy | NR | C | 7 (7; NR) | Brainstorming \| Cooperative learning \| Role playing \| Pair work \| Reflection \| Debate \| Case study | SWLS \| OLS |
| Choi 2016 [66] | 23 (NR) | Korea | Nursing | E | Self-esteem | NR | EC | 3 (7.5; NR) | Conferences | RSES |
| Choi et al. 2015 [67] | 87 (86%) | Korea | Nursing | QE | Communication \| Emotional intelligence | Salovey & Mayer 1990 [43] | EC | 8 (NR; weekly) | Lecture & presentations \| Group work \| Role playing \| Audio-visual \| Discussions | GICC \| AEQT |
| Christiansen & Jensen 2008 [68] | 288 (NR) | Norway | Nursing | QU | Interpersonal skills | NR | EC | 2 (NR; daily) | Role playing \| Case study \| Group work \| Debriefing | Focus group interviews \| Observation |
| Cole et al. 2023 [69] | NC = 22 (77%) SIC = 15 (80%) | USA | Pharmacy | QE | Self-awareness \| Conflict management \| Empathy | NR | NR | NR (NR; NR) | Audio-visual \| Lectures \| Discussions | EILI |
| Cunico et al. 2012 [70] | 103 (76%) | Italy | Nursing | QE | Empathy | NR | C | NR (21; NR) | Audio-visual \| Individual & couple exercises \| Discussions with lectures \| Role playing \| Observation \| Debriefing | BEES |

*(Continued)*

| Study | Sample size (% fem.) | Country | Degree programme | Study design | SECs promoted | Reported SECs theoretical framework | Modality | Nº of sessions (hours; frequency) | Teaching strategies | SECs measures |
|---|---|---|---|---|---|---|---|---|---|---|
| DasGupta & Charon 2004 [71] | 8 (100%) | USA | Medicine | QU | Empathy | NR | C | 5 & 6 (NR; weekly) | Writings | Reflective writing |
| Donisi et al. 2022 [72] | 264 (56%) | Italy | Medicine | QE | Communication \| Emotion handling skills \| Personal awareness | NR | C | NR (16; NR) | Interactive teaching \| Theoretical \| Audio-visual | IRI \| EQ-i |
| Duran et al. 2021 [73] | 30 (NR) | USA | Medicine | E | Communication | Frankel & Stein 1999 [74] | EC | NR (NR; NR) | Audio-visual \| Role playing \| Feedback | CARE-EC |
| Farver et al. 2016 [75] | 81 (33%) | USA | Medicine | QU | Self-awareness \| Self-management \| Social awareness \| Relationship management | Goleman 1995 [76] and Goleman et al. 2002 [77] | EC | 2 (NR; daily) | Readings \| Discussions | Skills survey \| Interviews |
| Fincias & González 2021 [78] | 30 (80%) | Spain | Social education | MM | Emotional competencies \| Self-regulation and emotional management \| Self-motivation \| Communication \| Empathy \| Assertiveness | NR | NR | 6 (15; weekly) | Active participation \| Role playing \| Debates \| Case study \| Audio-visual \| Readings | CDE-A35 \| Reflective writing |
| Fletcher et al. 2009 [79] | 70 (64%) | United Kingdom | Medicine | QE | Communication \| Emotional awareness | Bar-On 1997 [57] | EC | 7 (28; monthly) | Individual & grouped exercises \| Problem-based learning | EQ-i |
| Galal et al. 2012 [80] | 212 (67%) | USA | Pharmacy | QE | Communication \| Consideration of others \| Connection to others \| Self-awareness | Seal & Andrews-Brown 2010 [81] | C | NR (NR; weekly) | Simulations | SED-I |
| Gorgas et al. 2015 [82] | 33 (48%) | USA | Medicine | E | Social awareness | Kahn 2013 [83] | EC | 2 (4; NR) | Lectures \| Audio-visual \| Case study \| Group work | ECI-II |
| Goroshit & Hen 2012 [84] | 606 (85%) | Israel | Social work | QE | Self-awareness \| Interpersonal awareness \| Empathy | Mayer et al. 2000 [85] | C | NR (NR; weekly) | Experiential learning \| Journaling | SSREI |
| Goudarzian et al. 2019 [86] | 60 (53%) | Iran | Nursing | E | Self-awareness skills \| Empathy \| Management of excitement \| Problem-solving ability \| Decision-making skills \| Improve interaction with others \| Control anger \| Stress management | NR | EC | 12 (12; weekly) | Group work \| Lectures \| Personal and group reflections | EIA |
| Gunasingha et al. 2023 [87] | 108 (38%) | USA | Medicine | QE | Awareness of conflict resolution style | NR | NR | 1 (NR; once) | Simulations \| Feedback \| Audio-visuals | TKI |

*(Continued)*

**Table 2.** (Continued)

| Study | Sample size (% fem.) | Country | Degree programme | Study design | SECs promoted | Reported SECs theoretical framework | Modality | N° of sessions (hours; frequency) | Teaching strategies | SECs measures |
|---|---|---|---|---|---|---|---|---|---|---|
| Hen & Goroshit 2011 [88] | 165 (84%) | Israel | Social work | QE | Identify one's own emotions \| Express emotions and feelings \| Regulation of emotions \| Understand self-emotions and others | Mayer et al. 2000 [85] | C | NR (NR; weekly) | Discussions \| Pair work \| Group work \| Role playing \| Audio-visual \| Lectures \| Case study | SSREI IRI |
| Hurley et al. 2020 [89] | 12 (67%) | Australia | Nursing | QU | Recognize and manage emotions in themselves and others | Palmer et al. 2008 [90] | EC | 1 (4; once) | Feedback \| Coaching | GENOS EI Inventory \| Interview |
| Imperato & Strano-Paul 2021 [91] | 285 (NR) | USA | Medicine | QE | Emotional intelligence \| Empathy | NR | C | NR (NR; NR) | Reflection rounds \| Group work \| Feedback \| Discussion \| Case study | JSPE Student version \| WLEIS |
| Jiménez-Rodríguez et al. 2022 [92] | 60 (73%) | Spain | Nursing | QE | Understanding and management of emotions | NR | C | NR (NR; NR) | Audio-visual \| Case study \| Group work \| Role playing \| Individual work \| Homework | TMMS-24 |
| Kneese et al. 2020 [93] | 86 (57%) | USA | Medicine | MM | Express emotions \| Listening skills \| Witnessing: understanding and empathizing | Shapiro et al. 2006 [94] | EC | NR (NR; biweekly) | Discussions \| Anonymously paired participants \| Writings | Numerical Rating Scale \|Open questions |
| LeBlanc et al. 2017 [95] | 75 (64%) | United Kingdom | Psychology | E | Emotion regulation \| Stress regulation | NR | EC | NR (8; weekly) | Writings \| Mindfulness | ERQ \| PANAS |
| Lefroy et al. 2011 [96] | 245 (59%) | United Kingdom | Medicine | MM | Communication skills \| Empathy \| Emotional level control | NR | C | 5 (15; NR) | Simulations \| Group work | Ad hoc questionnaire \| Focus group |
| Lim et al. 2010 [97] | 40 (NR) | Korea | Nursing | E | Self-esteem | NR | EC | 8 (8; unclear) | Discussion \| Lectures | RSES |
| Lust & Moore 2006 [98] | 107 (NR) | USA | Pharmacy | MM | Empathy \| Conflict management | Goleman 1995 [99] | EC | NR (NR; NR) | Audio-visual \| Simulations \| Debriefing | Reflective questions and student feedback \| Essay questions |
| McConville & Lane 2006 [100] | 145 (NR) | NR | Nursing | QE | Self-efficacy to effectively communicate with patients | NR | EC | NR (NR; unclear) | Lectures \| Discussions | Self-efficacy toward nursing scale (ad hoc) |
| Mortari 2015 [101] | 39 (NR) | Italy | Faculty of Education | QU | Identify and comprehend self-emotions | NR | EC | NR (NR; NR) | Emotional journaling \| Conceptual analysis | Interviews |
| Orak et al. 2016 [102] | 33 (55%) | Iran | Nursing | QE | Emotion regulation \| Communication \| Assertiveness \| Self-awareness \| Empathy \| Problem solving \| Stress, anger and negative mood management | Salovey & Mayer 1990 [43] | C | 2 (16; weekly) | Lectures \| Role playing \| Brainstorming \| Homework \| Group teaching | MSSREI |

*(Continued)*

| Study | Sample size (% fem.) | Country | Degree programme | Study design | SECs promoted | Reported SECs theoretical framework | Modality | Nº of sessions (hours; frequency) | Teaching strategies | SECs measures |
|---|---|---|---|---|---|---|---|---|---|---|
| Pades-Jiménez et al. 2023 [103] | 753 (67%) | Spain | Physio-therapy \| Nursing \| Psychology | QE | Assertiveness \| Emotional intelligence \| Social skills | NR | C | NR (NR; NR) | Self/peer-assessment \| Reflection activities \| Modelling | Questionnaire \| TMMS-24 |
| Parks et al. 2019 [104] | 31 (77%) | USA | Physicians \| Dentists \| Public health practitioners | QE | Interpersonal skills | NR | C | NR (NR; NR) | Mentoring | EQSAC |
| Puffer et al. 2021 [105] | 75 (57%) | USA | Nursing | QE | Perception of emotions \| Facilitation of emotions \| Understanding of emotions \| Regulation of emotions | Salovey & Mayer 1990 [43] | EC | 1 (1; once) | Lectures \| Discussions \| Quiz | MSCEIT V2.0 |
| Raatikainen et al. 2022 [106] | 20 (NR) | Finland | Social work | QU | Empathy | NR | NR | NR (NR; NR) | Readings \| Lectures \| Personal reflection \| Discussions | Learning diaries \| Written documentation \| Open questionnaires |
| Redondo-Rodríguez et al. 2023 [107] | 102 (76%) | Spain | Psychology | QE | Emotional intelligence | NR | C | 3 (4; weekly) | Gamification \| Pair work \| Inter-disciplinary work | TMMS-24 |
| Reshetnikov et al. 2020 [108] | 242 (NR) | Russia | Medicine | QE | Self-motivation \| Empathy \| Ability to analyse oneself emotions and of others \| Emotional awareness \| Interpersonal relationships | NR | C | NR (276; weekly) | Case study \| Round tables \| Business games | Nicholas Hall Scale |
| Ribeiro et al. 2020 [109] | 74 (NR) | Brazil | Nursing | QE | Self-esteem \| Self-awareness Self-confidence \| Self-efficacy | NR | EC | 10 (NR; weekly) | Online through Facebook | RSES \| GSES |
| Rosa et al. 2015 [4] | NR (NR) | Spain | Social work \| Social education | MM | Conflict management \| Interpersonal skills | NR | C | NR (NR; weekly) | Coaching \| Interviews \| Debriefing | Focus group \| Ad hoc questionnaire |
| Rowland et al. 2019 [110] | 137 (76%) | Germany | Psychology | E | Self-control | NR | EC | 5 (NR; weekly) | Mindfulness meditation \| Homework | SCS \| SSCCS |
| Seow et al. 2022 [111] | 34 (56%) | NR | Medicine \| Dentistry \| Pharmacy \| Psychology | QE | Emotional intelligence | NR | NR | 4 (6; weekly) | Discussions \| Readings \| Experiential sharing | WLEIS |
| Shrivastava et al. 2022 [112] | 61 (NR) | NR | Medicine \| Pharmacy \| Physicians \| Psychology | MM | Emotional intelligence \| Communication skills | Interprofessional Education Collaborative 2016 [113] | EC | 1 (2.5; once) | Lectures \| Pair work \| Discussions \| Problem - solving \| Debriefing | SEL Toolkit \| Communication style inventory \| Quantitative and qualitative items |

*(Continued)*

**Table 2.** (Continued)

| Study | Sample size (% fem.) | Country | Degree pro-gramme | Study design | SECs promoted | Reported SECs theoretical frame-work | Modality | Nº of sessions (hours; frequency) | Teaching strategies | SECs measures |
|---|---|---|---|---|---|---|---|---|---|---|
| Sousa & Padovani 2021 [114] | 25 (96%) | Brazil | Psychology \| Nutrition \| Occupational Therapy | MM | Assertiveness \| Self-knowledge | NR | EC | 10 (10; weekly) | NR | ISS \| LSS \| PANAS \| Written evaluation |
| Watford & Stafford 2015 [115] | 70 (73%) | USA | Psychology | E | Experience and regulation of emotions | NR | EC | NR (NR; NR) | Mindfulness meditation | DERS-36 \| PANAS \| Heart Rate Variability \| Electroc. \| Skin conductance levels \| Electroenc. |

*Legend.* Fem. = Females; SECs = Socio-emotional competencies; NC = National cohort; SIC = Single institutional cohort; NR = Not reported; USA = United States of America; QU = Qualitative; QE = Quasi-experimental; E = Experimental; MM = Mixed-Methods; C = Curricular; EC = Extracurricular; AEQT = Adult Emotional Quotient Test; BES = Balanced Emotional Empathy Scale; CARE-EC = CARE Empathy Checklist; CDE-A36 = Emotional Development Questionnaire for Adults; DERS = Difficulties in Emotion Regulation Scale; ECCS = Empathic Communication Coding System; ECI-II = Hay 360 EI Quiz; EIA = Bradberry and Greaves' Standard Emotional Intelligence Questionnaire; EILI = Emotionally Intelligent Leadership Inventory; EQ-i = Emotional Quotient Inventory; EQSAC = Sterrett's Emotional Quotient Self-Assessment Checklist; ERQ = Emotion Regulation Questionnaire; ES = Empathy Scale; GICC = Global Interpersonal Communication Competence Scale; GRS = Global Rating Scale; GSES = General Self-Efficacy Scale; IRI = Interpersonal Reactivity Index; ISS = Inventory of Social Skills; JSPE = Jefferson Scale of Physician Empathy; LSS = Life Satisfaction Scale; MSCEIT = Mayer-Salovey-Caruso Emotional Intelligence Test; MSSREI = Modified Schutte Emotional Intelligence Scale; OGBHS = Observation Grid of the Basic Helping Skills; OLS = Overall Life Satisfaction Scale; PANAS = Positive Affect Negative Affect Scale; RSES = Rosenberg Self-Esteem Scale; SCS = Self-Control Scale; SED-I = Social Emotional Development Inventory; SEL Toolkit = Emotional Intelligence Self-Assessment Tools; SI = Strathclyde Inventory; SSCCS = State Self-Control Capacity Scale; SSREI = Schutte Self-Report Emotional Intelligence Test; SWLS = Satisfaction with Life Scale; TCCA-B = Test of the Basic Helping Skills (Test de Connaissances des Compétences d'Aide de Base, TCCA-B); TEIQue = Trait Emotional Intelligence Questionnaire; TICA = Interactive Test of the Helping Skills (Test Interactif des Compétences d'Aide, TICA); TKI = Thomas-Kilmann Conflict Mode Instrument; TMMS-24 = Trait Meta-Mood Scale; WLEIS = Wong and Law Emotional Intelligence Scale; Electroc. = Electrocardiogram; Electroenc. = Electroencephalogram.

Nearly half of the studies (n = 30; 47%) implemented extracurricular interventions, while 37% (n = 27) included them in the curriculum. The remaining 11% (n = 7) did not report this information. Furthermore, Fig 5 shows a classification of the different teaching/learning strategies employed in the intervention programmes to promote SECs, according to their curricular or extracurricular modality of delivery. In both cases, group work and debates/discussions stood out. In curricular interventions, case studies and role plays were also frequent, while, in extracurricular interventions, lectures, mindfulness/meditation classes and role plays were also common.

**3.4.3. Efficacy and effectiveness of intervention programmes.** The majority of reviewed intervention studies (n = 60, 94% of the total of intervention studies; 89% of curricular interventions, and 97% of the extracurricular interventions) reported significant changes in the measured variables of their target SECs, with only four studies not meeting the criterion of significance. None of the studies reported on the differential effectiveness of the various teaching and learning strategies used.

Nonetheless, gender emerged as a significant moderator variable of SECs in 4 out of the 64 intervention studies, predominantly showing favourable outcomes for women, who had higher post-intervention empathy scores [70], greater reductions in emotional worry [72], and higher EI scores over time [59,60], compared to men. The academic year and the university degree also influenced SECs outcomes in three studies. Advanced-year students were seen to have higher EI [88] and greater emotional clarity [103], with distinct improvements observed in third-year postgraduate medical residents

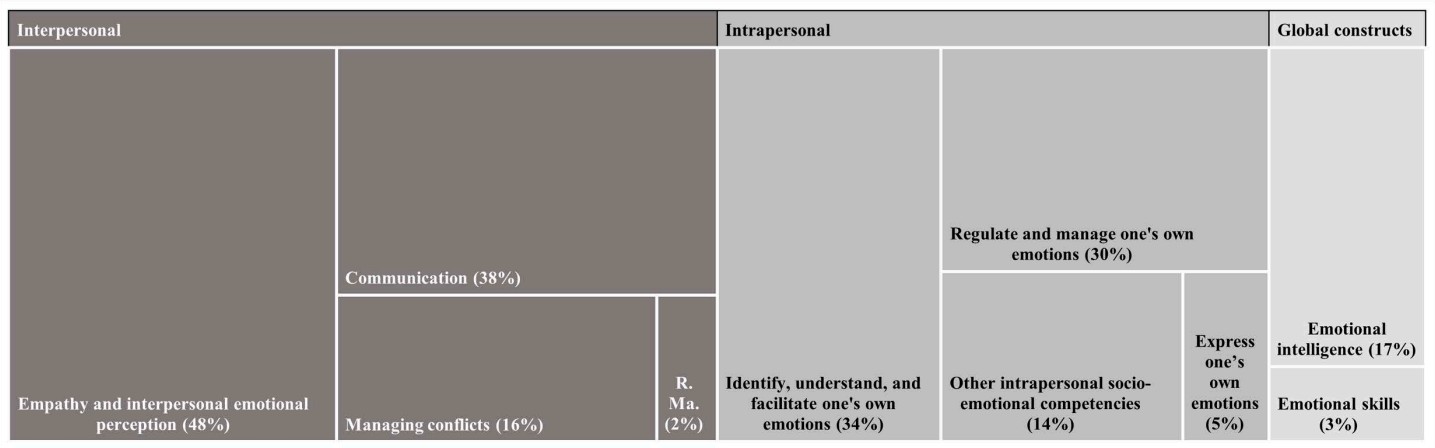

**Fig 4. SECs promoted in intervention studies.** *Legend. Empathy and interpersonal emotional perception*: recognize, understand, and manage others' emotions, respond sensitively and effectively, connect with others, and demonstrate social awareness and consideration; *Communication*: effective interaction with others, including skills such as active listening and assertiveness; *Managing conflicts*: effective management of disagreements, understanding different resolution styles and applying appropriate strategies to achieve constructive solutions; *R. Ma.*: relationship management, inspirational leadership, influence, teamwork, collaboration, and change catalyst; *Identify, understand, and facilitate one's own emotions*: emotional self-awareness, and recognition and understanding of one's own emotions to guide emotional responses in a constructive way; *Regulate and manage one's own emotions*: effective control one's emotions, to reduce stress, manage excitement, and apply self-control and self-management in emotional situations; *Other intrapersonal socio-emotional competencies*: self-esteem, self-efficacy, self-confidence, self-development, self-motivation, and decision-making; *Express one's own emotions*: communicating one's feelings and emotional states; *Emotional intelligence*: broad approach aimed at improving various SECs, without detailing specific competencies; *Emotional skills*: broad approach aimed at improving various SECs, without detailing specific skills.

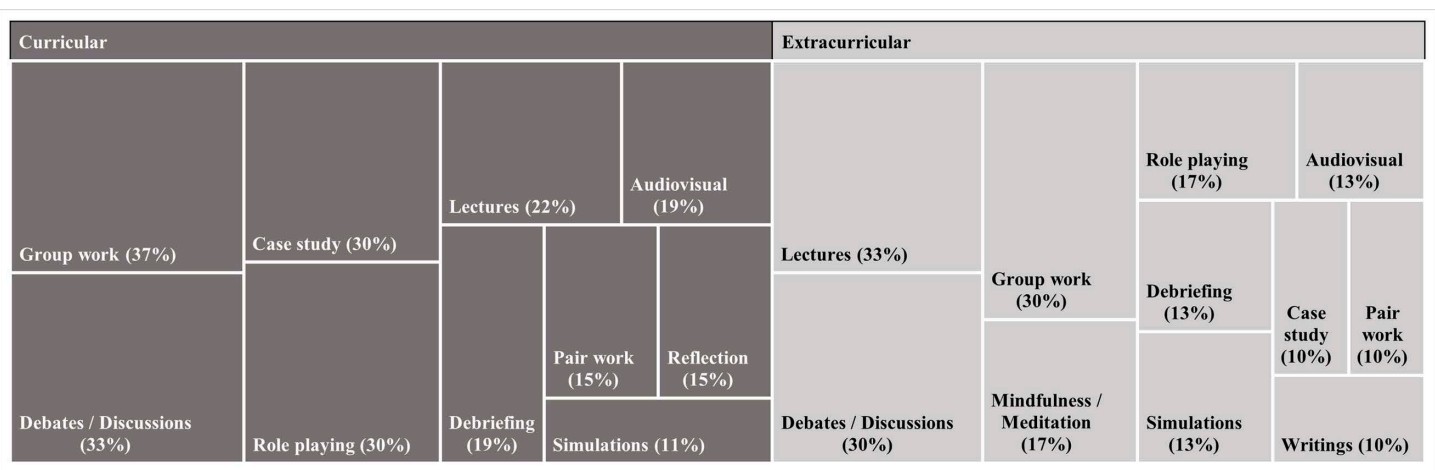

**Fig 5. Teaching/learning strategies by modality of delivery (curricular vs. extracurricular).** *Legend.* Figure created with those teaching/learning strategies with counts greater than 2 (n>2) or representing 10% or more. These teaching/learning strategies include: animation techniques, bibliotherapy, brainstorming, coaching, cooperative learning, experiential learning, gamification, journaling, mentoring, modelling, psychodrama, readings, storytelling and writing.

[60]. Regarding university degrees, Social Work degree students benefited more from an EI course than Education degree students [84], students in diagnostic specialities had higher negotiation and emotional quotient scores than those in procedural specialities [87], and Physiotherapy students achieved better emotional repair scores than Psychology students [103]. Other identified moderators included nationality, attendance, and ethnicity, highlighted in one study each [59,60,104].

## 4. Discussion

This scoping review identifies and synthesises existing empirical evidence on the strategies proposed to date to assess and promote university students' SECs in the socio-educational and healthcare fields. It has achieved two main objectives, examining in detail both the assessment of the SECs and the intervention programmes proposed for their promotion in both fields. More specifically, our scoping review provides: a) a compilation and ranking of the measures used to assess SECs; b) a classification of the most promoted SECs; and c) an analysis of the characteristics of the intervention strategies for their promotion at university level.

This in-depth analysis examines both the socio-educational and healthcare fields, and identifies two significant gaps in previous reviews. In the healthcare field, research has tended to focus on disciplines such as Nursing or Medicine, overlooking areas such as Psychology or Physiotherapy, and specifically excluding the socio-educational field. One of the most notable findings of this review is the disparity in the attention given to the assessment and promotion of SECs between the two sectors, with most studies coming from the healthcare field.

While the need to promote and assess SECs among professionals is evident in both sectors, the limited attention on the socio-educational field can be attributed to contextual differences and the unique challenges its professionals face. Healthcare professionals often deal with serious illness, pain management and even death within clinical settings [116], whereas socio-educational professionals focus on issues such as inequality and marginalisation, and typically work in community-based settings [117]. These professional and contextual differences may explain why the scientific community has not yet paid as much attention to the promotion and assessment of SECs among socio-educational professionals compared to those in healthcare. However, socio-educational professionals, such as social workers or social educators, must establish supportive and trusting relationships with their users. This fact requires strong interpersonal competencies, such as empathy, active listening, or assertiveness [118], as well as intrapersonal skills, such as emotional awareness and regulation [119]. For example, an educator working with at-risk young people who encounters a conflict needs to identify emotions, support emotional regulation, promote effective communication, and apply effective conflict resolution strategies. The use of these SECs is crucial in preventing the escalation of conflict and promoting the development of SECs in young people, enabling them to manage their emotions and resolve differences constructively. Future research would benefit from addressing this gap by focusing on the specific challenges faced by these professionals and the importance of applying SECs in their daily professional practice to improve outcomes for both the individuals they support and their own personal well-being.

### 4.1. SECs measures

An important aspect that emerges from our findings of this review is the complexity of assessing SECs in university education, both in the socio-educational and healthcare fields. Although it is considered a key element, it is a difficult task because SECs refer to a number of different competencies [11]. This is reflected in the variety of theoretical perspectives on the topic [12] and in the wide range of assessment tools [11,120].

Regarding the measurement tools used to assess SECs in university students in the socio-educational and healthcare field, it should be noted that the most commonly used standardised measures (i.e., SSREI, TEIQue, and MSCEIT) [40,44,46] are similar to those used for the general population [120]. Although these measures are primarily based on the construct of EI, they differ in the specific SECs they target. Therefore, according to Schoon [11], it is essential that the tools used to assess SECs are linked to the competencies defined in the theoretical framework of reference [121]. These tools are probably the most widely used, as they align with the most established theoretical frameworks in the field, thereby ensuring their reliability and validity in measuring SECs.

Another related aspect worth discussing is that we have identified a considerable number of non-specific and mixed measures that are currently used to assess SECs, even though they were not originally designed for this purpose. The use of such instruments to assess SECs may introduce several biases and research implications, including the risk that

these tools fail to capture the specific effects of interventions on SECs, as they are not designed to measure these competencies directly. Following O'Connor et al. [121], we recommend that researchers use measures of SECs that are most appropriate for their purpose.

The findings of our review are consistent with those of O'Connor et al. [121] who showed that there is no consensus on how to evaluate SECs. This fact presents a challenge in developing a consistent framework for their assessment and highlights the need for more specific and contextualised approaches [122,123]. Assessment tools should be standardised and adapted to students' profiles, taking into account their future roles and professional needs [124]. The SECs assessed should have a theoretical basis [121], clear definitions and be operationalised in several subdomains, as suggested by Berg et al. [12] and Schoon [11]. Furthermore, we believe that in intervention studies, the mixed-methods approach should be prioritised as the best methodological practice. This approach provides a more comprehensive understanding of SECs outcomes by integrating both quantitative data, which ensures statistical validity, and qualitative insights, which provides depth and context to the findings [125,126].

### 4.2. Intervention programmes to promote SECs

**4.2.1. SECs promoted in intervention programmes.** Throughout our review, we have also identified several challenges and gaps in the reporting of intervention programmes to promote SECs among university students. These findings are consistent with other reviews [34,127], which have similarly highlighted a lack of reporting and clarity around the theoretical frameworks used. According to Blyth et al. [128], establishing a theoretical framework for interventions is crucial as it provides a solid foundation for defining, organising, and classifying information, effectively communicating evidence, and focusing. It also aligns and mobilizes specific efforts to achieve the expected outcomes.

However, the diversity of theoretical frameworks found in the literature also reflects both the growing interest in the topic and the challenge of establishing a common language to unify the wide range of SECs. As noted by Berg et al. [12], Blyth et al. [128], and Schoon [11], this diversity complicates the development of a unified understanding of SECs. With this in mind, and according to our results, we propose a classification to organise the most commonly promoted SECs among university students in socio-educational and healthcare fields (see Fig 4). Our classification shows that empathy and interpersonal emotional perception were the most promoted SECs in intervention programmes. In line with this, the literature on socio-educational and healthcare professionals highlights that increased empathy improves user service adherence, satisfaction and working relationships, and reduces levels of anxiety and burnout among professionals [7]. Similarly, communication skills were among the most targeted SECs. Developing communication skills serves to more effectively gather information about people's needs through active listening and to more effectively provide feedback, information, education and necessary information to patients [72], while also providing physical and emotional support [129].

In the intrapersonal category, identifying, understanding and regulating one's own emotions was the third most promoted SEC. Professionals in socio-educational and healthcare settings are often faced with high-stress situations that can trigger intense emotional responses [119]. Effective emotional regulation helps these professionals make decisions and provide empathetic, patient-centred care [89]. Furthermore, understanding one's own emotions promotes resilience and prevents burnout [23], ensuring that workers can sustain their performance and commitment in the long term.

Given the complexity and importance of these competencies, we believe that intervention programmes should focus exclusively on promoting SECs. However, less than half of the intervention studies reviewed took this focused approach, with many also addressing other factors, such as ego identity [66], creative thinking [65], or professional identity and narrative competence [93]. While these factors are valuable, their inclusion may dilute the primary focus on SECs, potentially limiting the effectiveness of interventions in achieving their intended outcomes.

**4.2.2. Characteristics of intervention programmes.** Although most of the intervention studies reviewed show general effectiveness [79,130], few have identified significant moderator variables. Identifying moderator variables allows researchers and practitioners to understand how different factors—such as gender, academic year, age, or work

experience—might influence the effectiveness of SECs interventions [28,131]. Furthermore, recognising these factors can also help in customising tailored interventions to meet the specific needs of different subgroups, thereby improving their overall effectiveness [132]. Although there is evidence supporting the importance of addressing these gaps, more concrete recommendations are needed on how to approach them. For example, to address gender differences, we suggest that interventions be designed flexibly, incorporating specific learning strategies or adjusting the frequency and intensity of activities. Regarding the academic year, we propose that students in the early stages of their education may require a more structured and guided approach, focused on knowledge acquisition and basic skills training. In contrast, those in more advanced stages might benefit from more autonomous interventions, using teaching strategies such as those based on simulations or real-life situations during their curricular practices.

Another result to be highlighted is that intervention modalities reveals no significant differences between curricular and extracurricular approaches, with both being effective in promoting SECs [66,133], suggesting flexibility in the implementation of the intervention. However, we believe that integrating SECs into curricular training is valuable as it positions these competencies as a mandatory, common, central and ongoing aspect of students' education, rather than a supplementary or temporary focus [134]. This integration also ensures the sustainability of the interventions by embedding them within the educational framework [135], using existing resources rather than relying on additional ones. It can also offer several additional benefits, such as increasing student engagement and motivation, which in turn has a positive impact on academic performance and overall well-being [136,137].

The Bologna Declaration [138] and the European Association for Quality Assurance in Higher Education [139] highlight the importance of combining academic knowledge with transferable skills. This combination may facilitate the personal and professional development of students, while emphasising the need for the continuous improvement of curricula to ensure they remain up-to-date and responsive to changing societal and disciplinary demands. In this context, the ECTS Users' Guide, published by the European Association of Institutions in Higher Education [140], establishes that all degree programmes encompass both generic and specific competencies. The latter refer to those that involve the demonstration of knowledge, skills, and abilities in both work and academic contexts, with responsibility and autonomy. In this regard, we consider SECs to be specific competencies that could be integrated into curricula in socio-educational and healthcare fields. We propose integrating them as elective courses, which could be worth around 3 ECTS [140], depending on the flexibility of each university's curriculum. Additionally, we believe it would be beneficial to offer this course before or alongside the internship period. This approach would enable students to apply and refine these competencies in real-world settings, optimizing their integration and practical development.

On a different note, we identified that the predominant teaching/learning strategies were group work, debates and discussions, and lectures. However, no significant differences in outcomes were reported based on the teaching strategy used. Research in this area would be valuable in identifying which strategies lead to the most successful outcomes in the acquisition of SECs in our target population. Given the complexity of the subject, each SEC may require a different approach [141]. Existing literature focuses primarily on specific SECs, such as empathy, or more generally on the construct of EI, and is largely drawn from the health field. Effective strategies highlighted in this literature include simulation [33,127,142], role-playing [142], experiential immersive learning [33], and games and discussions [127].

As previously stated, and in line with the findings from other SECs intervention reviews [33,132], there is a lack of detailed procedural reporting. Examples of missing information include the number of sessions, intervention duration and other relevant factors. This inadequate reporting, combining with the significant heterogeneity observed across studies, presents a challenge to the replicability of studies and makes it difficult to compare their results. In light of these findings, we recommend that future research adopt reporting standards for interventions based on intervention design, such as the CONSORT [143] or TIDieR [144] guidelines. In particular, this information should include the following aspects of the implementation of educational interventions: a) integration into curricular or extracurricular activities; b) timing within the course or curriculum; c) experience/expertise required of instructors and prerequisites for the intervention context; d)

scheduling of the educational intervention, including the number of sessions, their frequency, timing and duration, and the amount of time learners spend in self-directed learning activities; e) the method of delivery and the learner/teacher ratio; f) the teaching/learning strategies applied; g) the support materials used; h) the expected effects of each component of the intervention and specific outcome measures; i) the facilitators and barriers to the implementation of the intervention; j) any specific adaptations planned; k) any unplanned changes during the implementation of the intervention; and l) any potential moderating variables.

Furthermore, we suggest that future studies prioritise evaluating the impact of key elements of the intervention process, to better understand their influence on SECs outcomes.

**4.2.3. Additional methodological recommendations to improve intervention programmes.** In light of our results, we emphasise the importance of tailoring the specific SECs targeted by intervention programmes to the students' profile, taking into account their needs in relation to their future professional roles [28]. This adaptation can increase student engagement and motivation, as students are more likely to perceive a direct and significant impact on their career goals [136]. Second, the measures used to assess SECs should be standardised and specifically tailored to the SECs being targeted, in line with the theoretical framework of the intervention [12]. As mentioned above, a mixed-methods approach is recommended to gain a deeper understanding of the outcomes of interventions aimed at promoting SECs [125,126]. In addition, conducting long-term measurements allows for the assessment of the lasting effects of interventions, helping to identify both permanent and transient outcomes that may not be evident in the short term [145]. Additionally, they facilitate the identification of moderating variables that may influence the effectiveness of interventions and provide important data for optimising future evidence-based interventions.

Finally, regarding the quality of intervention studies, it is essential to address their complex nature by ensuring comprehensive reporting. In order to facilitate the replication of studies and a thorough review of their effects on the target SECs, detailed information on their procedures must be adequately reported [31,127,146].

## 4.3. Study limitations and strengths

The results of this scoping review should be interpreted in the context of several limitations. Some of these limitations are inherent to this type of study. For example, there is a potential for publication bias due to the over-representation of certain types of studies. These include studies that confirm the authors' hypotheses or whose results are statistically significant. To mitigate this bias, a comprehensive search was conducted across multidisciplinary, health and education databases, as well as grey literature, to ensure the review's comprehensiveness. Another limitation of this study is the exclusive inclusion of studies written in English and Spanish, which may have introduced bias in the literature selection. However, the review ensures broad geographical coverage. For example, although studies in Portuguese were excluded, the review includes several investigations conducted in this region [15,147,148]. Further limitations include the challenges encountered in extracting key data from the studies, including details on the duration of the intervention, procedures, timeframe for outcome assessment, reliability and validity of measures, and data on the socio-demographic aspects of the sample. Due to the limited number of studies assessing certain subgroups of data, additional planned comparisons could not be carried out. Furthermore, difficulties were encountered in synthesising the heterogeneous results, particularly with regard to SECs measures. The inclusion of a wide range of SECs measures in the studies, coupled with a lack of standardisation in fundamental aspects such as the measure name and its dimensions, presented a significant challenge. In light of these limitations, it is advisable to exercise caution when generalising the results of the scoping review.

Notwithstanding the limitations, in our opinion, this scoping review has important strengths, stemming from an updated and comprehensive search strategy, that represents a significant improvement over previous review. Thus, the large number of included studies is representative of diverse geographical regions (e.g., America, Europe, and Asia) and university programmes (e.g., Nursing, Psychology, Physiotherapy, or Social Work). Furthermore, the review included a wide range of study designs and objectives. It did not focus exclusively on studies related to the development or adaptation of

measures, but also included descriptive, correlational, and experimental studies. Another key strength is the inclusion of both healthcare and socio-educational settings, as this review is the only one to date to extend the scope of SECs studies beyond both settings.

## 5. Conclusions

There is a clear consensus in the literature that SECs are highly needed and valued in all socio-educational and healthcare fields, although they have been studied more extensively in the health sector.

In terms of how SECs are assessed in university students, there is a wide variety of measurement tools, stemming from different theoretical perspectives on SECs. In addition, the use of non-specific or mixed measures that were not originally designed to assess SECs is evident, highlighting the need for future research to validate their effectiveness. In this regard, we suggest the development and validation of assessment tools that are aligned with a robust theoretical framework and adapted to the profiles of university students based on their roles and future professional needs.

Regarding intervention programmes, less than half of the studies focused solely on promoting SECs, often aiming to improve other competencies simultaneously. Based on our findings, the most commonly promoted SECs included empathy and interpersonal emotional perception, communication, and the identifying, understanding and regulating of one's own emotions. The application of these competencies has been demonstrated to yield positive outcomes in several areas, including the enhancement of user relationships, the facilitation of sound decision-making processes and the mitigation of professional burnout. While intervention programmes have demonstrated effectiveness in general, further research is needed on the impact of moderating variables and other key characteristics of the intervention procedure on SEC outcomes. A clear theoretical framework is also essential for guiding the design and implementation of these programmes. Additionally, it is crucial that the targeted SECs align with the specific characteristics and demands of the university degree to ensure their efficacy. Lastly, it should also be emphasised that integrating the assessment and promotion of SECs into university curricula is critical to better prepare helping professionals to provide more holistic care and support to individuals.

## Supporting information

**S1 Table. Main theoretical frameworks of SECs.**
(DOCX)

**S2 Table. Search strategy of each database.**
(DOCX)

**S3 Table. Data extraction protocol.**
(DOCX)

**S1 Checklist. PRISMA-ScR checklist.**
(DOCX)

**S1 Text. References of included studies.**
(DOCX)

**Dataset 1. Dataset of measurement studies.**
(XLSX)

**Dataset 2. Dataset of descriptive and correlational studies.**
(XLSX)

**Dataset 3. Dataset of intervention studies.**
(XLSX)

**Dataset 4. Dataset of measures reviewed.**
(XLSX)

**Dataset 5. Dataset of SECs promoted in intervention studies.**
(XLSX)

## Author contributions

**Conceptualization:** Natalia Gandia Carbonell, Paz Viguer, Josep-Maria Losilla.

**Data curation:** Natalia Gandia Carbonell, Cristian Molla-Esparza, Sònia Lorente, Paz Viguer, Josep-Maria Losilla.

**Formal analysis:** Natalia Gandia Carbonell, Cristian Molla-Esparza, Paz Viguer, Josep-Maria Losilla.

**Funding acquisition:** Josep-Maria Losilla.

**Investigation:** Natalia Gandia Carbonell.

**Methodology:** Natalia Gandia Carbonell, Cristian Molla-Esparza, Sònia Lorente, Paz Viguer, Josep-Maria Losilla.

**Resources:** Natalia Gandia Carbonell, Cristian Molla-Esparza, Sònia Lorente, Paz Viguer, Josep-Maria Losilla.

**Software:** Natalia Gandia Carbonell.

**Supervision:** Paz Viguer, Josep-Maria Losilla.

**Validation:** Natalia Gandia Carbonell, Paz Viguer, Josep-Maria Losilla.

**Visualization:** Natalia Gandia Carbonell, Paz Viguer, Josep-Maria Losilla.

**Writing – original draft:** Natalia Gandia Carbonell, Sònia Lorente, Paz Viguer, Josep-Maria Losilla.

**Writing – review & editing:** Natalia Gandia Carbonell, Cristian Molla-Esparza, Paz Viguer, Josep-Maria Losilla.

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
