## [Decision Letter · Decision Letter 0]

24 Feb 2025

PONE-D-24-56078Strategies to assess and promote the socio-emotional competencies of university students in the socio-educational and healthcare fields: A scoping reviewPLOS ONE

Dear Dr. Gandia Carbonell,

Thank you for submitting your manuscript to PLOS ONE. After careful consideration, we feel that it has merit but does not fully meet PLOS ONE’s publication criteria as it currently stands. Therefore, we invite you to submit a revised version of the manuscript that addresses the points raised during the review process.

**Address the comments and suggestions given by the reviewers and Academic Editors.**

We look forward to receiving your revised manuscript.

Kind regards,

Jordan Llego, PhD ELM, D. Hon. Ex., PhDN, RN

Academic Editor

PLOS ONE

**Journal Requirements:**

1. When submitting your revision, we need you to address these additional requirements. Please ensure that your manuscript meets PLOS ONE's style requirements, including those for file naming. The PLOS ONE style templates can be found at https://journals.plos.org/plosone/s/file?id=wjVg/PLOSOne_formatting_sample_main_body.pdf and https://journals.plos.org/plosone/s/file?id=ba62/PLOSOne_formatting_sample_title_authors_affiliations.pdf 2. Thank you for stating in your Funding Statement:  The research is supported by Grant PID2022-141403NB-I00, from the Spanish Ministry of Science and Innovation. Sonia Lorente Sànchez has been awarded the Margarita Salas Fellowship, funded by the European Union, Next-Generation EU, in collaboration with the Ministry of Universities of Spain, and the Autonomous University of Barcelona.Please provide an amended statement that declares *all* the funding or sources of support (whether external or internal to your organization) received during this study, as detailed online in our guide for authors at http://journals.plos.org/plosone/s/submit-now.  Please also include the statement “There was no additional external funding received for this study.” in your updated Funding Statement. Please include your amended Funding Statement within your cover letter. We will change the online submission form on your behalf. 3. Please note that your Data Availability Statement is currently missing the repository name. If your manuscript is accepted for publication, you will be asked to provide these details on a very short timeline. We therefore suggest that you provide this information now, though we will not hold up the peer review process if you are unable. 4. Please include captions for your Supporting Information files at the end of your manuscript, and update any in-text citations to match accordingly. Please see our Supporting Information guidelines for more information: http://journals.plos.org/plosone/s/supporting-information.

**Additional Editor Comments:**

To enhance your manuscript, it is essential to incorporate the suggestions provided by the reviewers. Additionally, here are my suggestions:

First, you should expand the discussion on the practical implications of Social-Emotional Competencies (SECs), focusing on how the findings can inform curriculum design in healthcare and socio-educational fields. Additionally, it explores how SEC training can be institutionalized within higher education policies to ensure its integration into academic frameworks.

Next, consider enhancing the comparative analysis of SEC measurement tools by including a table outlining key assessment tools' psychometric properties, such as validity and reliability. It would also be beneficial to clarify which tools are most commonly used and why they are popular.

Addressing the socio-educational representation gap is another critical area to strengthen. Develop a more structured argument explaining why the socio-educational field has lagged in SEC research and propose potential policy recommendations or funding priorities that could help bridge this gap.

Furthermore, in the section on intervention studies, a systematic review of best practices could be improved by creating a matrix that categorizes interventions based on their effectiveness, target competencies, and methodologies employed. Lastly, it is important to strengthen the methodology section by providing greater transparency in data extraction and synthesis processes to enhance reproducibility. Additionally, explicitly acknowledging potential biases in literature selection—such as language and publication biases—will strengthen the manuscript's credibility.

Reviewers' comments:

Reviewer's Responses to Questions

**Comments to the Author**

1. Is the manuscript technically sound, and do the data support the conclusions?

Reviewer #1: Yes

Reviewer #2: Yes

2. Has the statistical analysis been performed appropriately and rigorously? 

Reviewer #1: N/A

Reviewer #2: N/A

3. Have the authors made all data underlying the findings in their manuscript fully available?

Reviewer #1: Yes

Reviewer #2: Yes

4. Is the manuscript presented in an intelligible fashion and written in standard English?

Reviewer #1: Yes

Reviewer #2: Yes

5. Review Comments to the Author

**Reviewer #1:**  I would like to express my appreciation to the authors for their thorough and insightful scoping review on the assessment and promotion of socio-emotional competencies (SECs) among university students in the socio-educational and healthcare fields. This is a highly relevant topic, given the increasing recognition of SECs as essential skills for professionals in these disciplines. The authors have undertaken a significant effort in mapping the existing empirical literature, identifying gaps, and providing a structured synthesis of key findings. Their work contributes to an important and evolving field, offering valuable insights that may inform future research, educational policies, and intervention designs.

Strengths of the Manuscript

Comprehensive Scope and Methodological Rigor

The manuscript follows a systematic and transparent approach, adhering to the PRISMA-ScR guidelines. The methodology is well-detailed, with clear inclusion and exclusion criteria, ensuring replicability.

The use of multiple databases and a grey literature search enhances the comprehensiveness of the review.

Clear Structuring and Thematic Organization

The paper effectively categorizes studies into measurement studies, descriptive/correlational studies, and intervention studies, providing a structured approach to the field.

The distinction between intra-personal and inter-personal SECs is valuable and aids in conceptual clarity.

Identification of Research Gaps

The review highlights significant gaps in the literature, particularly regarding SECs assessment tools, the underrepresentation of socio-educational fields compared to healthcare, and the lack of standardized measures.

The discussion on inconsistencies in SECs definitions and theoretical frameworks is important for advancing the field.

Constructive Recommendations for Future Research

The authors make well-founded recommendations regarding the need for standardized and context-specific SECs measures.

The suggestion to integrate SECs training within university curricula is timely and aligns with current trends in higher education.

Areas for Improvement and Suggested Revisions

Clarification of Theoretical Frameworks

While the manuscript mentions various theoretical models of SECs, there is a need for a clearer synthesis of these frameworks to guide readers. A concise table summarizing the key models and their respective SECs dimensions would be helpful.

Greater Emphasis on the Socio-Educational Field

The review acknowledges the dominance of healthcare studies in SECs research, yet it would benefit from a more detailed discussion on why this gap exists and how future research can better address it.

More specific examples of the role SECs play in socio-educational professions would strengthen this section.

Intervention Studies: Standardization and Longitudinal Analysis

The paper highlights the heterogeneity of SECs intervention studies but could provide a more structured analysis of best practices. A comparison table outlining the most effective intervention strategies and their outcomes would be beneficial.

The need for longitudinal studies to assess the long-term impact of SECs training is briefly mentioned but should be emphasized further.

Limitations and Reporting Transparency

Some details regarding study selection and data extraction processes could be expanded to further ensure transparency and reproducibility.

Given the reliance on published studies, potential publication bias should be acknowledged more explicitly, with suggestions for addressing this issue in future research.

Language and Readability

The manuscript is well-written but could benefit from minor refinements to enhance clarity and readability. Some sections are dense with technical details; breaking up long paragraphs or simplifying certain explanations could improve reader engagement.

**Reviewer #2:**  Thank you very much for reviewing this article. Below are the following suggestions for improvement:

On page 13, lines 273–280, the study notes the heterogeneity of instruments used to assess social and educational competencies. The authors could expand their discussion on the need to develop and adopt standardized measures specifically designed to assess SEC.

On page 20, lines 345–347, it appears that 69% of intervention studies are identified, limiting the replicability of the theoretical framework. The authors could emphasize this as a recommendation in the discussion in line with established frameworks.

Page 21, lines 374–389: Many intervention studies lack information on the number, duration, and frequency of sessions. The authors should recommend improved reporting standards for future intervention research. and highlight the need for better intervention reporting guidelines.

Page 22, Lines 405-416: The variables, such as gender and academic year, are identified as moderators in some intervention studies. The discussion could explore the implications of these findings more thoroughly, particularly how interventions could be tailored based on these factors.

In figure 4, ensure that captions are detailed and self-explanatory for standalone readability and would benefit from a brief summary of the categories in their captions.

Maintain consistent use of terms such as “social and emotional competencies” and “emotional intelligence” throughout the manuscript to avoid confusing the reader.

The authors state that they follow the PRISMA-ScR guidelines; it would be nice to include the checklist in the supplementary materials for confirmation.

6. PLOS authors have the option to publish the peer review history of their article (what does this mean? ). If published, this will include your full peer review and any attached files.

**Do you want your identity to be public for this peer review?** For information about this choice, including consent withdrawal, please see our Privacy Policy .

Reviewer #1: **Yes: ** Nicola Pagnucci

Senior Researcher in Nursing Science

PhD, MSN, RN, FFNMRCSI

Department of Translational Research and of New Surgical and Medical Technologies

University of Pisa

Reviewer #2: No

---

## [Author Response · Author response to Decision Letter 0]

3 Apr 2025

We would like to thank you and the reviewers for providing valuable feedback on how to improve the manuscript. We greatly appreciate your consideration and the reviewers’ comments, and we have implemented the proposed changes as suggested.

Below you will find our point-by-point responses to each issue raised by the reviewers. Every change listed in this letter is also present in the revised manuscript. The changes made in the manuscript have been indicated in purple so that the editor and reviewers can easily locate them.

Yours sincerely.

Editor:

1. First, you should expand the discussion on the practical implications of Social-Emotional Competencies (SECs), focusing on how the findings can inform curriculum design in healthcare and socio-educational fields. Additionally, it explores how SEC training can be institutionalized within higher education policies to ensure its integration into academic frameworks.

Authors’ response. Thanks for your note. In lines 581 to 597, we discuss how SECs can be integrated into curricula in the socio-educational and healthcare fields. Based on frameworks such as the Bologna Declaration (1999) and the European Association for Quality Assurance in Higher Education (2015), we highlight the importance of combining academic knowledge with transferable skills to enhance students' personal and professional development. Additionally, we propose a specific approach for their curricular incorporation.

2. Next, consider enhancing the comparative analysis of SEC measurement tools by including a table outlining key assessment tools' psychometric properties such as validity and reliability. It would also be beneficial to clarify which tools are most commonly used and why they are popular.

Authors’ response. We appreciate your suggestion. A column has been added to Table 1 detailing the number of studies that report adequate validity and reliability indices. Additionally, in the Datasets 1-3 you can find the specific indices for each instrument from all studies that report them. Lastly, in lines 489 to 492 we have provided an explanation of why the mentioned measures are among the most commonly used.

3. Addressing the socio-educational representation gap is another critical area to strengthen. Develop a more structured argument explaining why the socio-educational field has lagged in SEC research and propose potential policy recommendations or funding priorities that could help bridge this gap.

Authors’ response. We agree. In lines 454 to 462, we have already provided an explanation of why this gap may exist in SEC research within the socio-educational field. However, we have expanded the argument further in lines 462 to 475, offering a more structured discussion and a specific example of the need for SECs in the socio-educational field.

4. Furthermore, in the section on intervention studies, a systematic review of best practices could be improved by creating a matrix that categorizes interventions based on their effectiveness, target competencies, and methodologies employed.

Authors’ response. Thank you for your suggestion. Upon considering your recommendation, we realised that the studies we examined do not provide detailed strategies for each SEC, which limits the possibility of constructing such a matrix. The aim of this scoping review is to identify and describe the intervention programs developed to promote SECs at the university level, focusing on their theoretical frameworks, target SECs, and outcomes. Given that our primary goal is to map the landscape of SEC interventions, rather than to establish a best practices framework, the creation of such a matrix falls beyond the scope of this review. However, we do present findings on the effectiveness and efficiency of the interventions in lines 419 to 437, and specific data on these aspects can be found in column V of Dataset 3.

5. Lastly, it is important to strengthen the methodology section by providing greater transparency in data extraction and synthesis processes to enhance reproducibility. Additionally, explicitly acknowledging potential biases in literature selection—such as language and publication biases—will strengthen the manuscript's credibility.

Authors’ response. Thank you for pointing this out. On the one hand, as described in lines 173 to 174, data extraction was performed according to a predefined coding protocol. Additionally, to enhance transparency and reproducibility, we have added a supplementary table (S3 Table) that provides a detailed description of the variables and their corresponding definitions. On the other hand, in lines 657 to 661, we explicitly acknowledge the potential biases in literature selection.

Reviewer 1:

1. Clarification of Theoretical Frameworks. While the manuscript mentions various theoretical models of SECs, there is a need for a clearer synthesis of these frameworks to guide readers. A concise table summarizing the key models and their respective SECs dimensions would be helpful.

Authors’ response. We appreciate your valuable suggestion. To provide clearer guidance to readers, we have included Table S1, which summarizes the key theoretical models and their respective SECs.

2. Greater Emphasis on the Socio-Educational Field. The review acknowledges the dominance of healthcare studies in SECs research, yet it would benefit from a more detailed discussion on why this gap exists and how future research can better address it. More specific examples of the role SECs play in socio-educational professions would strengthen this section.

Authors’ response. We have already addressed this valuable contribution as previous mentioned in Editor Comment 3. In lines 454 to 462, we have already provided an explanation of why this gap may exist in SEC research within the socio-educational field. However, we have expanded the argument further in lines 462 to 475, offering a more structured discussion and a specific example of the need for SECs in the socio-educational field.

3. Intervention Studies: Standardization and Longitudinal Analysis. The paper highlights the heterogeneity of SECs intervention studies but could provide a more structured analysis of best practices. A comparison table outlining the most effective intervention strategies and their outcomes would be beneficial.

Authors’ response. Thank you for your suggestion, as previous mentioned in Editor Comment 4, upon considering your recommendation, we realised that the studies we examined do not provide detailed strategies for each SEC, which limits the possibility of constructing such a matrix. The aim of this scoping review is to identify and describe the intervention programs developed to promote SECs at the university level, focusing on their theoretical frameworks, target SECs, and outcomes. Given that our primary goal is to map the landscape of SEC interventions, rather than to establish a best practices framework, the creation of such a matrix falls beyond the scope of this review. However, we do present findings on the effectiveness and efficiency of the interventions in lines 419 to 437, and specific data on these aspects can be found in column V of Dataset 3.

4. The need for longitudinal studies to assess the long-term impact of SECs training is briefly mentioned but should be emphasized further.

Authors’ response. According to your comment, the importance of longitudinal studies to assess the long-term impact of SEC training has been further emphasized in lines 640 to 645.

5. Limitations and Reporting Transparency Some details regarding study selection and data extraction processes could be expanded to further ensure transparency and reproducibility. Given the reliance on published studies, potential publication bias should be acknowledged more explicitly, with suggestions for addressing this issue in future research.

Authors’ response. Firstly, as previous mentioned in Editor Comment 5, we have added a table (S3 Table) to enhance transparency and reproducibility. This table provides a detailed description of the variables and their corresponding definitions. Secondly, in lines 608 to 629, we explicitly address potential publication bias by citing two reference guidelines for tackling this issue and proposing specific elements that need to be reported in future research to mitigate this bias.

6. Language and Readability. The manuscript is well-written but could benefit from minor refinements to enhance clarity and readability. Some sections are dense with technical details; breaking up long paragraphs or simplifying certain explanations could improve reader engagement.

Authors’ response. Thank you for your valuable feedback. We have taken your suggestion into account and made several adjustments throughout the manuscript. Some of the modifications can be seen in the Introduction (lines 67 to 71; 84 to 87), Methodology (lines 183 to 184), Results (lines 234 to 235; 315 to 319) or Discussion (lines 677 to 680).

Reviewer 2:

1. On page 13, lines 273–280, the study notes the heterogeneity of instruments used to assess social and educational competencies. The authors could expand their discussion on the need to develop and adopt standardized measures specifically designed to assess SECs.

Authors’ response. We appreciate your input on this point. In lines 497 to 498, the discussion has been expanded to address the biases associated with measuring SECs using instruments not specifically designed for this purpose.

2. On page 20, lines 345–347, it appears that 69% of intervention studies are identified, limiting the replicability of the theoretical framework. The authors could emphasize this as a recommendation in the discussion in line with established frameworks.

Authors’ response. According to your comment, this point has already been addressed in lines 519 to 522, where we emphasize the importance of establishing a theoretical framework for interventions, as it provides a structured foundation for organizing information, communicating evidence, and guiding actions to achieve expected outcomes.

3. Page 21, lines 374–389: Many intervention studies lack information on the number, duration, and frequency of sessions. The authors should recommend improved reporting standards for future intervention research and highlight the need for better intervention reporting guidelines.

Authors’ response. In agreement with your comment, in lines 608 to 629 we have added recommendations for future research to adopt standardized reporting guideline. Moreover, we explicitly outline key aspects of intervention implementation, including integration into curricular activities, scheduling details, instructor expertise, delivery methods, and expected outcomes.

4. Page 22, Lines 405-416: The variables, such as gender and academic year, are identified as moderators in some intervention studies. The discussion could explore the implications of these findings more thoroughly, particularly how interventions could be tailored based on these factors.

Authors’ response. We appreciate your suggestion. We have expanded the discussion in lines 560 to 569 to further explore how interventions could be tailored considering gender and academic year as moderating variables. This addition highlights the potential implications of these factors in designing more effective.

5. In figure 4, ensure that captions are detailed and self-explanatory for standalone readability and would benefit from a brief summary of the categories in their captions.

Authors’ response. Thank you for pointing this out. The caption for Figure 4 has been revised to include detailed descriptions of each category, ensuring that it is clear and comprehensible on its own.

6. Maintain consistent use of terms such as “social and emotional competencies” and “emotional intelligence” throughout the manuscript to avoid confusing the reader.

Authors’ response. Thank you for your observation. We have ensured consistent use of the term social and emotional competencies throughout the manuscript. The term emotional intelligence only appears when referring to specific theoretical frameworks or measurement tools.

7. The authors state that they follow the PRISMA-ScR guidelines; it would be nice to include the checklist in the supplementary materials for confirmation.

Authors’ response. The PRISMA-ScR checklist has been included in the supplementary S1 Checklist, where it can be reviewed for confirmation. Please let us know if any additional details are required.

---

## [Decision Letter · Decision Letter 1]

27 Apr 2025

Strategies to assess and promote the socio-emotional competencies of university students in the socio-educational and healthcare fields: A scoping review

PONE-D-24-56078R1

Dear Dr. Gandia Carbonell,

We’re pleased to inform you that your manuscript has been judged scientifically suitable for publication and will be formally accepted for publication once it meets all outstanding technical requirements.

Kind regards,

Jordan Llego, PhD ELM, D. Hon. Ex., PhDN, RN

Academic Editor

PLOS ONE

Additional Editor Comments (optional):

Thank you for submitting the revised version of your manuscript to PLOS ONE. After careful evaluation of your detailed responses and the updated manuscript, I am pleased to inform you that your article is accepted for publication.

The revised manuscript has substantially addressed all editorial and reviewer comments. You have notably enhanced the clarity, methodological rigor, and transparency of your work.

Overall, your scoping review offers a valuable and timely contribution to the literature by mapping and analyzing strategies to assess and promote SECs among university students in both socio-educational and healthcare fields. Your work will serve as a useful foundation for future research and educational program development.

On behalf of the PLOS ONE editorial team, I congratulate you and your co-authors on your excellent work. Your manuscript will now proceed to production. You will be contacted by the production team regarding the proofing process.

Thank you for choosing PLOS ONE as the venue for your research. We look forward to seeing your work published and reaching the broader academic community.

Reviewers' comments:

Reviewer's Responses to Questions

**Comments to the Author**

1. If the authors have adequately addressed your comments raised in a previous round of review and you feel that this manuscript is now acceptable for publication, you may indicate that here to bypass the “Comments to the Author” section, enter your conflict of interest statement in the “Confidential to Editor” section, and submit your "Accept" recommendation.

Reviewer #2: All comments have been addressed

2. Is the manuscript technically sound, and do the data support the conclusions?

Reviewer #2: Yes

3. Has the statistical analysis been performed appropriately and rigorously? 

Reviewer #2: N/A

4. Have the authors made all data underlying the findings in their manuscript fully available?

Reviewer #2: Yes

5. Is the manuscript presented in an intelligible fashion and written in standard English?

Reviewer #2: Yes

6. Review Comments to the Author

Reviewer #2: Thank you for your thoughtful and comprehensive revision of the manuscript entitled "Strategies to assess and promote the socio-emotional competencies of university students in the socio-educational and healthcare fields: A scoping review."

After carefully reviewing the revised version and your point-by-point responses, I confirm that all major comments raised by the editor and reviewers have been fully addressed, and the manuscript has significantly improved in clarity, depth, and methodological rigor.

1. Your explanation (pp. 17–18, lines 454–475) now provides a structured rationale for the underrepresentation of socio-educational fields in SEC research. For that, you might consider, in future work, a focused follow-up review or empirical study targeting this specific gap.

2. You addressed concerns about tool variability in SEC measurement (p. 15, lines 489–492; p. 16, lines 497–498). I support your emphasis on the need for validated, SEC-specific instruments. This point is well-justified in your updated Table 1 and related discussion to fully support standardization of measures and reporting.

3. The rationale for not providing a best-practices matrix is clearly explained (p. 13–14, lines 419–437), and you effectively redirected the focus to general trends in effectiveness. Your inclusion of detailed intervention data in Dataset 3 adds valuable depth.

4. The expansion on how gender and academic year may influence SEC outcomes is appreciated (p. 20, lines 560–569). This enhances the manuscript’s practical relevance for tailoring interventions in future studies.

7. PLOS authors have the option to publish the peer review history of their article (what does this mean? ). If published, this will include your full peer review and any attached files.

**Do you want your identity to be public for this peer review?** For information about this choice, including consent withdrawal, please see our Privacy Policy .

Reviewer #2: No

---

## [Editor Report · Acceptance letter]

PONE-D-24-56078R1

PLOS ONE

Dear Dr. Gandia Carbonell,

I'm pleased to inform you that your manuscript has been deemed suitable for publication in PLOS ONE. Congratulations! Your manuscript is now being handed over to our production team.

Kind regards,

on behalf of

Dr. PLOS Manuscript Reassignment

Staff Editor

PLOS ONE